# Influencing Factors of Acceptance and Use Behavior of Mobile Health Application Users: Systematic Review

**DOI:** 10.3390/healthcare9030357

**Published:** 2021-03-22

**Authors:** Chen Wang, Huiying Qi

**Affiliations:** Department of Health Informatics and Management, School of Health Humanities, Peking University, Beijing 100191, China; wangchenparis@bjmu.edu.cn

**Keywords:** users’ behavior, individual influencing factors, society influencing factors, App design influencing factors

## Abstract

Purpose/Significance: Mobile health applications provide a convenient way for users to obtain health information and services. Studying the factors that influence users’ acceptance and use of mobile health applications (apps or Apps) will help to improve users’ actual usage behavior. Method/Process: Based on the literature review method and using the Web of Science core database as the data source, this paper summarizes the relevant research results regarding the influencing factors of the acceptance and use behavior of mobile health application users and makes a systematic review of the influencing factors from the perspectives of the individual, society, and application (app or App) design. Result/Conclusion: In terms of the individual dimension, the users’ behavior is influenced by demographic characteristics and motivations. Social attributes, source credibility, and legal issues all affect user behavior in the social dimension. In the application design dimension, functionality, perceived ease of use and usefulness, security, and cost are the main factors. At the end of the paper, suggestions are given to improve the users’ acceptability of mobile health applications and improve their use behavior.

## 1. Introduction

With socio-economic development and the improvement in human living standards, people are paying more attention to health, and their pursuit of health has gradually changed from treating disease to preventing disease, which leads to the increasing demand for portable medical care. With the rapid development of mobile Internet technology and the popularity of intelligent terminal mobile devices, mobile health applications have exploded as the most direct tools for public personal health management.

Mobile health applications, the principal manifestation of mobile healthcare, refer to health applications based on mobile terminal systems such as Android and iOS that provide services such as medical information inquiry and symptom self-examination. Mobile health applications allow users not only to seek answers to health problems but also to gain access to healthcare, exercise and fitness, health management, and other related services anytime and anywhere. Mobile health applications alleviate the shortage of health information resources to a certain extent, provide a convenient way for users to obtain health information and services, and play an important role in spreading health knowledge and meeting the users’ need for health consultation. It is pointed out in the Healthcare Mobile Application Market Size, Share & Trends Analysis Report By Type (Fitness Products Training, Appointment Booking & Construction), By Platform, By Technology, By End User, And Segment Forecasts, 2020–2027 that the global market size of healthcare mobile applications was valued at 17.92 billion US dollars in 2019, and it is expected to grow at a compound annual growth rate of 45% from 2020 to 2027 [1].

As a mobile health application provides users with a convenient and timely opportunity to acquire health knowledge and self-management [2], its market size is growing rapidly, and a large number of applications have been created to help people manage their health conditions, indicating that this type of application has gained global attention. However, no matter how the mobile health application (app or App) develops, the determining factor is always the users who enter the data. What are the factors that affect users’ acceptance and use of mobile health applications (apps or Apps)? In view of this problem, this paper systematically reviews the relevant research and finds that, although research results are abundant, most studies focus on the empirical research on the influencing factors of users’ acceptance and use of mobile health apps, including research on different groups of people, such as the determinants of the adoption of mobile health apps among college students [3], the analysis of mobile health app possession and related sociodemographic factors among Hong Kong citizens [4], and the factors associated with mobile health app use in cardiovascular diseases or diabetes patients [5]; research on mobile health apps with different functions, such as the acceptability of mental health apps [6], the opportunities and obstacles in mobile health app use for cancer care [7], and the relationship between the characteristics of mobile health apps related to maternal and infant health and users’ intent to use [8]; and research on a certain influencing factor, such as the effect of sending a push notification on engagement with a mobile health app [2], the relationship between health intentions and mobile health app ownership [9], and the obstacles and facilitators in using mobile health apps from the perspective of security [10]. In addition, previous reviews have mainly focused on a certain type of mobile health app (e.g., for patients with rheumatic and musculoskeletal diseases [11] or for chronic arthritis patients [12]). To the best of our knowledge, no study has systematically reviewed the influencing factors of users’ acceptance and use behavior of mobile health apps in order to reflect the general situation of current research. Therefore, this paper performs a comprehensive review to comb the research and analyze the influencing factors from multiple dimensions in order to grasp the current situation of and the trend in relevant research and provide a reference for follow-up research.

## 2. Materials and Methods

According to Bradford’s law, most of the key research is published in core international journals. Therefore, the Web of Science core database was taken as the data source in this paper, and we searched articles between 1 January 1900 and 7 June 2020. The search strategy was as follows: Topic= (“mobile health application” OR “mobile health applications” OR “health app” OR “health apps”) AND the literature type (Article); 772 papers were obtained. The selection process was conducted in accordance with the Preferred Reporting Items for Systematic Reviews and Meta Analyses extension for Scoping Reviews (PRISMA-ScR) [13] and is summarized in Figure 1. Any paper that met at least one of the following criteria was excluded: (1) duplication (*n* = 5); (2) non-English (*n* = 33); (3) irrelevant terms (e.g., influencing factors of users’ acceptance and use behavior of non-mobile health applications; *n* = 526); and (4) review articles (*n* = 5). A total of 203 papers remained in the final sample.

## 3. Results

### 3.1. Year of Publication

The number of studies is an important index to evaluate the attention, input, and influence of scholars in a certain field. The annual change of published articles can directly reflect the research status of the influencing factors of mobile health application users’ behavior. The temporal distribution of literature is shown in Figure 2. The earliest related literature appeared in 2009, in which the author explored people’s views and attitudes toward mobile health applications through a national opinion survey [14]. On the whole, the number of published papers showed a continuous growing trend from 2009 to 2019, especially since 2014, when the increase marked a rapid growth in papers. It can be indicated that research on the influencing factors of users’ acceptance and use has attracted more and more attention as the mobile health application industry has rapidly developed. In order to ensure research timeliness, this review also includes 25 research papers that were published in 2020 (before June).

### 3.2. Journal Distribution

The journals that have published at least three papers are shown in Table 1. It can be seen that the related research on the influencing factors of mobile health application users’ behavior mainly involves the fields of health care sciences and services, medical informatics, and computer science. The publishing of such articles started mainly after 2013.

### 3.3. Influencing Factors in Acceptance and Use Behavior of Mobile Health Application Users

Mobile health application users’ behavior is influenced by many factors, such as users themselves, other users, external environment, App functions, and App content. Therefore, this paper will summarize 10 influencing factors from three dimensions (the individual, society, and App design) for systematic review, and a spider plot according to the frequency of each influencing factor in the relevant literature is shown in Figure 3.

#### 3.3.1. Individual Dimension

From the individual dimension, the factors that affect users’ acceptance and use of mobile health applications can be summarized as demographic characteristics (such as age, gender, education level, and income) and motivations (such as health awareness and e-health literacy) (as shown in Table 2).

Demographic characteristics.

First of all, use behavior partly depends on age, as mobile health applications are used more by the young (although the definition for the age range of young people in various literature is different, for most it is under 35 years old [15,16,17,18]) rather than seniors, especially those over age 70, who are less comfortable in using their phone applications [15]. Some seniors do not even have mobile applications [16], and they tend to seek health information in traditional mass media. From the perspective of gender disparity, it is supported by lots of evidence that females have a higher tendency than males to utilize mobile devices to search and make use of health information [19]. However, in actual use, gender disparity is only obvious in mobile health applications with specific functions. Males prefer fitness applications, while females prefer applications related to nutrition, self-healthcare, and reproductive healthcare [20]. For example, research on the use of reproductive health applications among American college students reported that females showed more interest in this kind of application than males [21]. In terms of education level, users with higher education levels have a higher tendency to use mobile health applications [22] and better staying power. As for income, the possession of smartphones and the adoption of phone applications have a positive correlation with income status [23]. A lower income leads to the lower utilization of information technology related to mobile health [4]. Thus, income is one of the important predictors for users who use mobile devices to promote health [24]. Populations with a high level of medical insurance are more likely to adopt and use mobile health applications. When it comes to region, people in rural areas are less likely to use mobile health applications [16]. In research on veterans’ attitudes toward mobile health applications, compared to urban veterans, rural veterans reported that smartphones went against their values, and they tended to disagree with the usage of applications [25]. Moreover, the digital gap among ethnic groups still exists though it is narrowing [26]; difficulties in searching and understanding health information for non-native speakers may restrict their acceptance and use behavior of mobile health applications [27]. Lastly, for the health condition variable, those with a lower self-rated health status and a lack of physical exercise are less likely to adopt and use mobile health applications [28]. However, Shen C reported that having a history of chronic diseases was associated with the adoption of mobile health applications [4].

2.Motivation.

Motivation is an internal driving force that causes one to engage in certain activities. Health awareness, as one type of motivation, refers to the degree of concern for one’s own health. As the purpose of mobile health applications is to maintain or improve users’ health conditions, individual health awareness has a direct impact on the use of mobile health applications [29]. Thus, whether to accept and use mobile health applications may reflect whether users have the motivation to change or maintain health behavior [30]. For example, in the use of applications, those with the intention to change their weight differ from those who pay no attention to their current state [31]. The expression of a higher willingness to lose weight and do exercises is related to the adoption of mobile health applications [9]. Similarly, Milward J et al. found that abstainers were motivated to use an alcohol harm reduction application because they wanted to reduce alcohol consumption [32].

In addition, users are required to have the ability to accurately understand the information in mobile health applications [29], and e-health literacy represents the users’ ability to use information technology for seeking and adopting health information, which leads to the solution of health problems. Mobile health application users have a higher e-health literacy level than non-users. Users who believe that applications have a greater effectiveness on health behavior tend to know e-health better [5]. Therefore, Bol N et al. suggested that e-health literacy can be regarded as a prerequisite for the use of mobile health applications [20].

#### 3.3.2. Social Dimension

In addition to the individual dimension, users may also be affected by social environments and other users, as users are not isolated. The social influence on users’ acceptance and use behavior can be discussed from three aspects: social attributes, source credibility, and legal issues (as shown in Table 3).

Social attributes.

The sharing/social network function is integral for phone applications. This function enables users to share information and achievements and communicate with other users, which facilitates studying health lifestyles and insisting on health activities [33]. It is worth mentioning that via the sharing/social network function fitness applications convert individual exercise behavior into social behavior, make exercises more interesting, and promote the users’ sense of achievement, eventually resulting in the increment of users’ staying power.

2.Source credibility.

As mobile health applications provide a large amount of health information, users are often exposed to ambiguous or controversial messages. Given that such information has a great impact on health, the source credibility of mobile health applications is a concern for users. In general, traditional mass media (e.g., television, newspapers, and magazines) are regarded as reliable and credible sources of information [34]. Cho J found that the perceived credibility of health information from traditional mass media had a positive correlation with users’ cognition of mobile health applications. Reports of the application in traditional mass media may influence users’ views and attitudes toward mobile health applications [35]. Professionals (doctors, pharmacists, and nurses) are also an important source of medical information; thus, their recommendations are one of the influencing factors of users’ acceptance and use behavior. Health practitioners should provide instructions and suggestions for such applications and suggest the most suitable, reliable, explicable, and manageable applications for users. When they recommend the most reliable applications to users, they play an important role in ensuring the quality of applications [36]. Likewise, healthcare organizations or official medical institutions as the providers of mobile health applications, or the publicity and support of related organizations or institutions, help to enhance the credibility of mobile health applications, which is a strong determining factor in users’ acceptance and use behavior [37].

3.Legal issues.

Although increased credibility may influence users’ acceptance and use behavior, accompanying legal issues may become a hindrance to the use of applications. As information provision, collection, and transmission are highly dependent on virtual networks, mobile health applications are highly legally risky. It is difficult for users to determine the authenticity of the personal or health information of professionals in applications, and it is difficult to clarify legal responsibility in case of medical or drug safety issues. Lack of legal supervision reduces users’ trust and thus hinders the acceptance and use of mobile health applications [38]. In addition, privacy protection and security issues during the process of information collection and transmission are also of concern to users.

#### 3.3.3. App Design

The importance of the design characteristics of mobile health applications should not be underestimated. The App design should be attractive to users and meet their needs so that users can accept and use applications (as shown in Table 4).

Functionality.

The main function of mobile health applications is to provide users with medical and health information, and thus the quality of information is one of the key factors of users’ acceptance and use of applications. Information quality can be measured from three aspects: accuracy, timeliness, and relevance. Applications designed by non-medical institutions would result in concerns about the accuracy of information, thus affecting the use of applications [8]. Regular or timely updates have a positive effect on use [39]. Applications that enable users to communicate with health professionals, other users, or providers will improve the users’ healthcare experience [40].

Users’ interest in mobile health applications is also determined by other core functions (e.g., reminders, notifications, incentives, follow-up, and goal setting) and the way these functions are provided. For example, although they are provided in other devices, reminders, as a popular function among users, are very convenient in mobile applications. It should be noted that the timing and frequency of the reminder must be well designed or it will be ignored by users [41]. Sending push notifications containing customized health information is related to the engagement in mobile health applications. Bidargaddi N et al. found that users were most likely to use the application in 24 h when the notification was sent at noon at the weekend [2]. Incentive mechanisms such as virtual badges, levels unlocked, and behavior data comparison with other users are also considered as driving forces of use [41]. Many users like the follow-up function, as they think such a function can promote health awareness and observe the progress of health behavior; however, it may affect the user experience if not properly implemented [42]. The goal-setting function is also popular, as many users believe it promotes self-discipline and helps them to change their behavior gradually. Some users indicate that it may have a better effect if it was combined with follow-up, real-time feedback, and progress reports [41].

For users, personalization is a key characteristic that promotes attractiveness and acceptability. Different populations have various preferences in relation to the core functions of mobile health applications. For example, the reminder is necessary for drug trackers, while it may have the opposite effect in applications that help users to quit smoking [42]. Furthermore, many people have the motivation to carry out health activities but do not know the right way for themselves. Therefore, users believe that applications customized for them can satisfy their preferences and health management goals [43] and provide personalized guidance, enabling the use of applications to achieve highly personalized interventions and thus significantly promote users’ abilities to understand and manage their health conditions [44].

While the core functions mentioned above provide a reasonable basis for users’ acceptance and use, gamification gives an emotional support that maintains users’ motivations [45]. In applications involved with high levels of self-monitoring, gamification fosters users’ optimistic emotions and enthusiasm, makes the application more incentive-based and pleasant to use [46], and helps users to pursue their goals and promote health improvements such as running faster, eating healthier, and quitting smoking.

2.Perceived ease of use and usefulness.

In the Technology Acceptance Model proposed by Davis in 1989, perceived ease of use and perceived usefulness are the two main factors in users’ acceptance of information systems. Perceived ease of use refers to the labor-saving degree of use, while perceived usefulness refers to the degree in which use can improve personal performance. Research studies have shown that perceived ease of use and usefulness have a positive correlation with users’ willingness to accept and use mobile health applications [6,47,48,49].

Perceived ease of use in mobile health applications is mainly reflected in the user interface design and use efficiency. When the user interface is filled with text or information, it is difficult for users to interact with the application, while a clean and simple interface can help users to navigate it [50]. In addition, some specific needs for specific users, such as the button size and data visualization for seniors [51], are also factors to measure the friendliness of the mobile health application interface.

If the application can collect data automatically and promptly without users’ frequent input, and be compatible with other devices and platforms (e.g., blood glucose-related applications and blood glucose monitors (or blood glucose pumps) [52], mobile phones and tablet computers, and Apple and Android systems [6]), it will enhance the use efficiency and affect the degree of user utilization and the willingness to use it continuously [50]. However, healthcare practitioners believe that heavy offline workloads restrict their time and effort to learn, adapt to, and use the application, even if they approve of it [53]. Furthermore, using the application in addition to familiar workflows may be disruptive, and they are concerned about extra workloads. Unless the application is effective, time-saving, and easy to use, healthcare practitioners are unlikely to use it [54]. It can be indicated that having to invest an enormous amount of energy and time is a hindrance to users’ acceptance and use.

The perceived usefulness of mobile health applications is mainly reflected in providing relevant health information according to users’ particular situations, better managing their health, recording related data for users to review and observe, and satisfying the needs of social networking and sharing with others. These factors have been commented on in the information quality, core functions, personalization, and social attributes above and thus will not be elaborated on here.

3.Security.

In health data, security is always related to privacy, indicating that any unauthorized access to users’ health data (a security hole) is an invasion of privacy [10], and mobile health applications based on the Internet are more vulnerable to security attacks and interceptions. Users’ concern about the security and privacy of health data is one of the reasons that they do not use or do not continuously use mobile health applications. Health information is considered as absolutely private information [55], thus users have the right to decide whether to share it, conceal it, or share it anonymously [56]. In addition, some mobile health applications lack a privacy policy or lack clarity even if they have one [57]. It has been reported that most users want to know how to control access to their health data [10]. When users depend on mobile health applications to record and save data, they are concerned about who has access to these data and how the service provider will use them [58]. In addition, users express preferences for some security functions, such as periodic password updates, remote erases, informed consent, and access control [10]; thus application providers should clearly show these security and privacy functions to users.

4.Cost.

Cost, a common reason for users to reject adopting mobile health applications, includes the requirement of terminal devices, the cost of purchasing applications, and data traffic costs when using the applications. In the research conducted by Krebs P and Duncan DT, cost was the top concern of users. Most of them were unwilling to pay for mobile health applications, and nearly half of them stopped using the application because of hidden costs [22]. Free applications are likely to encourage users to use them in health management [10].

## 4. Discussion

Although many mobile health applications have been proved to help maintain or improve users’ health conditions, many people reject adopting them or use them only for a short period. This suggests that research on the influencing factors of users’ acceptance and use behavior of mobile health applications is still significantly challenging. Thus, based on the current research, the following suggestions are given.

### 4.1. Considering the Demographic Factors That Affect Users’ Behavior

Demographic characteristics can provide a reference for researchers and guide how to carry out mobile medical interventions for different audiences. By summarizing the demographic factors that influence users’ behavior, it has been found that the gap within age, education, income level, and health condition is large, and the populations in most need (e.g., seniors and populations with low health literacy) seldom use the service and receive little benefit from advanced medical healthcare information technology. In addition, users of mobile health applications around the world have their own cultural and language differences. Studies have shown that culture is an important component that contributes to the success of a system or product. Integrating certain cultural values of specific user groups into user interface designs may promote users’ acceptance. Thus, mobile health applications are required to provide pertinent and personalized designs that meet users’ cultural and language needs to enhance their utilization rate and further achieve wide participation and healthcare equity.

### 4.2. Strengthening the Publicity and System Management of Mobile Health Apps

With the spread of mobile devices and social networks, health administrations should pay more attention to the information technology related to health. Disseminating the advantage of correctly using mobile health applications may promote use, help users to manage their own health, and promote the early diagnosis and prevention of diseases. Meanwhile, the government should supervise applications strictly and improve the policies, laws, and regulations for the collection, analysis, and storage of individual health data.

### 4.3. Strengthening the Security and Privacy Measures of Mobile Health Apps

The security and privacy of the collection and analysis of data in mobile health applications greatly influence users’ perception, which may further determine their willingness to use such applications; therefore, application providers should recognize the importance of security and privacy policies and inform users clearly in order to ensure data security. On the other hand, mobile health applications should keep a balance between data security and providing more functions.

## 5. Limitations

This review was restricted to the article literature in the Web of Science core database and did not include other types of literature (such as scientific reports and white papers) or other literature databases (such as PubMed and Scopus); therefore, the research results may not have captured all the possible influencing factors of mobile health applications. Non-English articles were also excluded, which limits the generalizability of our findings.

## 6. Conclusions

Previous studies have mainly focused on finding and verifying the factors that affect the users’ acceptance and use behavior of mobile health applications. Different from previous research, this paper summarizes 10 influencing factors and 24 related variables from three dimensions (the individual, society, and App design), analyzes the importance of different factors according to the frequency of influencing factors in the relevant literature, and systematically reviews the influencing factors of users’ acceptance and use behavior of mobile health apps.

In the individual dimension, users’ behavior is influenced by demographic characteristics and motivation. Demographic characteristics including age, gender, education level, income level, medical insurance, region, ethnicity and language, and health condition have been studied by many researchers, while health awareness and e-health literacy are the main variables of motivation.

In the social dimension, social attributes, source credibility, and legal issues all affect user behavior. Specifically, source credibility mainly influences users’ acceptance and use behavior in three aspects: media publicity, recommendation by professionals, and support from healthcare institutions.

As for App design, the research findings are relatively rich, including functionality, perceived ease of use and usefulness, security, and cost. Firstly, the Technology Acceptance Model is widely used in studying the factors that affect users’ behavior, in which perceived ease of use and usefulness are the factors that appear most frequently in the relevant literature. Secondly, functionality is the focus of researchers, in which information quality is an important indicator; however, core functions (e.g., reminders, notifications, incentives, follow-up, and goal setting), personalization, and gamification also attract the attention of researchers. Thirdly, the importance of security cannot be ignored. Finally, research on cost provides a new perspective for application designers.

As the user’s intention and behavior play a key role in the innovation and development of mobile health apps, it is bound to be favored by more researchers in follow-up research. In the future, research may focus on the influencing factors in different application scenarios.

## Figures and Tables

**Figure 1 healthcare-09-00357-f001:**
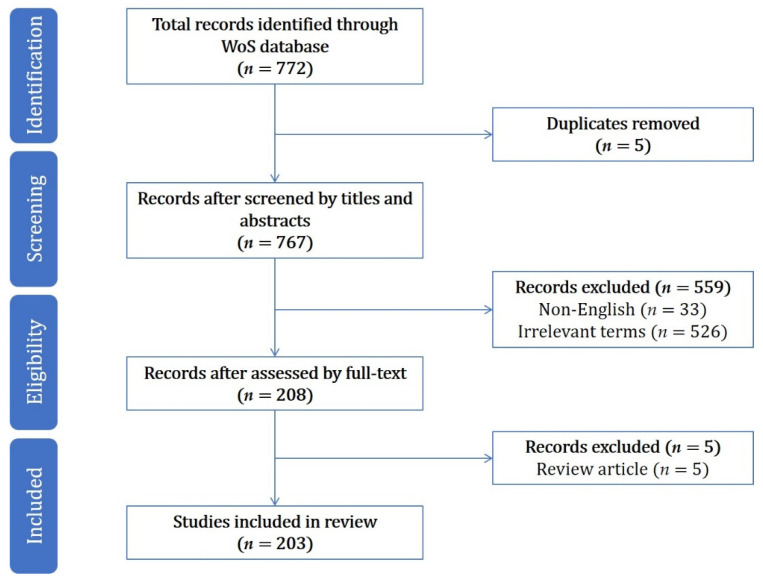
Article search and screening process.

**Figure 2 healthcare-09-00357-f002:**
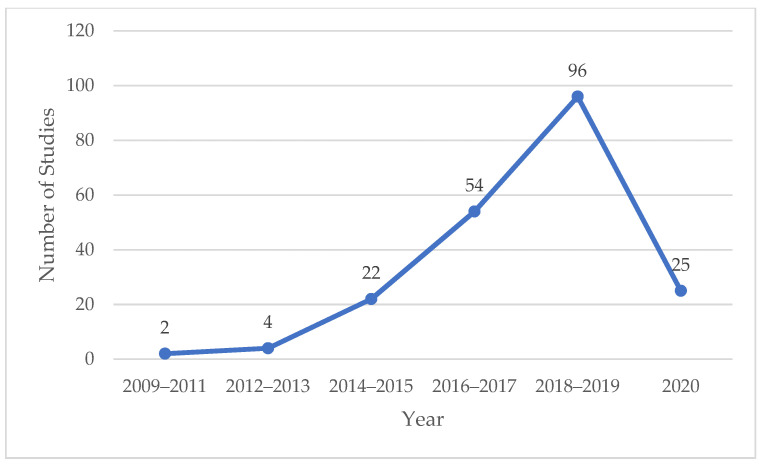
Temporal distribution of literature.

**Figure 3 healthcare-09-00357-f003:**
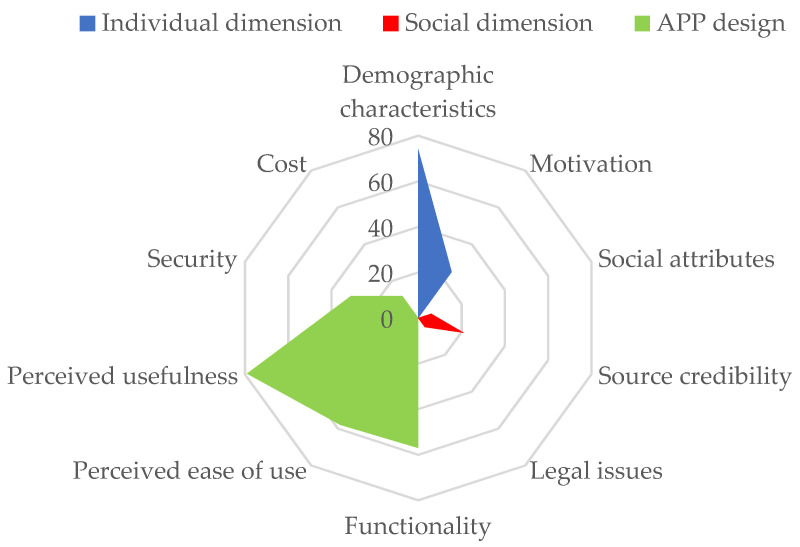
Influencing factors of acceptance and use behavior of mobile health apps.

**Table 1 healthcare-09-00357-t001:** Journal distribution.

Journals	Number of Studies	Journal Fields	Start Publishing Papers
JMIR mHealth and uHealth	7	Health Care Sciences & Services	2014
International Journal of Medical Informatics	6	Health Care Sciences & Services, Medical Informatics, Computer Science	2009
Journal of Medical Internet Research	5	Health Care Sciences & Services, Medical Informatics	2014
BMC Medical Informatics and Decision Making	4	Medical Informatics	2013
PLOS One	4	Comprehensive	2016
Telemedicine and E-Health	3	Health Care Sciences & Services	2015
Health Informatics Journal	3	Health Care Sciences & Services	2016
Digital Health	3	Health Care Sciences & Services	2018
Journal of Cancer Education	3	Medicine Education	2018

**Table 2 healthcare-09-00357-t002:** Influence of individual factor in use behavior.

Influencing Factor (Number of Related Papers)	Related Variable	Main Conclusion	Literature
Demographic characteristics (74)	Age	Mobile health applications are mainly used by the young, while seniors, especially those aged over 70, seldom adopt or use mobile health applications	[15,16,17,18]
	Gender	Males prefer fitness applications, while females prefer applications related to nutrition, self-healthcare, and reproduction	[19,20,21]
	Education level	Populations with a high education level tend to use mobile health applications	[22]
	Income	Income levels are positively correlated with the use of mobile health applications	[4,23,24]
	Medical insurance	Populations with a high medical insurance level have a stronger tendency to use mobile health applications	[16]
	Region	Compared to urban residents, rural ones are not likely to use mobile health applications	[16,25]
	Ethnicity and language	Difficulties in searching and understanding health information for non-native speakers may restrict their acceptance and use	[26,27]
	Health condition	Those with a lower self-rated health status are less likely to use mobile health applications; a chronic history is associated with use	[4,28]
Motivation (25)	Health awareness	Populations with higher health awareness have a higher acceptance level of mobile health applications	[9,29,30,31,32]
	E-health literacy	E-health literacy is considered as a prerequisite of using mobile health applications	[5,20,29]

**Table 3 healthcare-09-00357-t003:** Influence of social factor in use behavior.

Influencing Factor (Number of Related Papers)	Related Variable	Main Conclusion	Literature
Social attributes (6)	Sharing/social network	Social networking increases interests and a sense of achievement and promotes users’ staying power	[33]
Source credibility (21)	Media publicity	Perceived credibility of traditional mass media has a positive correlation with the cognition of mobile health applications	[34,35]
	Recommendation of professionals	As an important source of medical information, healthcare professionals’ recommendations may influence users’ behavior	[36]
	Support from healthcare organization	Support from healthcare institutions may enhance the credibility of mobile health applications	[37]
Legal issues (5)	Legal supervision	Lack of legal supervision decreases the users’ trust of mobile health applications	[38]

**Table 4 healthcare-09-00357-t004:** Influence of App design in use behavior.

Influencing Factor (Number of Related Papers)	Related Variable	Main Conclusion	Literature
Functionality (57)	Quality of information	Accuracy, timeliness, and relevance positively affect users’ trust in mobile health applications	[8,39,40]
	Core function	Users are interested in core functions such as reminders, notifications, encouragement, follow-up, and goal setting, as well as in the way they are provided	[2,41,42]
	Personalization	Personalization is a key characteristic that enhances the attractiveness and acceptability of mobile health applications	[42,43,44]
	Game-based	Gamification provides an emotional support for maintaining motivations	[45,46]
Perceived ease of use (58)	User interface design	A clean and simple interface can help users better interact with applications	[50,51]
	Efficiency	The enormous amount of energy and time consumed is a hindrance to users’ acceptance and use	[6,50,52,53,54]
Perceived usefulness (79)	Perceived usefulness	This includes information quality, core functions, personalization, and social attributes	[2,8,33,39,40,41,42,43,44]
Security (31)	Security and privacy	Users’ concern about the security and privacy of health data is one of the reasons they do not use or do not continuously use mobile health applications	[10,55,56,57,58]
Cost (12)	Cost	Some users are not willing to pay for mobile health applications	[10,22]

## Data Availability

Not applicable.

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
