# Peer review of "Influencing Factors of Acceptance and Use Behavior of Mobile Health Application Users: Systematic Review"

_healthcare, 2021, doi:10.3390/healthcare9030357_

Round 1

Reviewer 1 Report

Mobile technology and electronic applications provide great services at the level of the individual and society on the one hand, and institutions and sectors (such as health) on the other hand. However, this technology is still not used by some individuals and institutions. There are many reasons and factors influencing not using this technology. Therefore, this topic is very important and attracts researchers to uncover these challenges. This research presented a systematic review in identifying factors affecting the acceptance of mobile health applications. The authors split these factors into individual, society and application design. They provided detailed information on this topic. The methodology is clear and the research structure is appropriate. Authors must address all of the following concerns carefully.

  • We suggest that the authors should eliminate keywords such as “mobile health applications”, “influencing factor” and “systematic review " because these keywords are already found in the article title. It is preferable that they replace them with other words in order to expand the reach of the search.
  • The abstract is appropriately written, structured, complete and independent as well as it is correctly describing the title of the paper. However, the authors must define what the acronym "APP" (page1-line7) indicates before using it.
  • The introduction section provides appropriate background on the topic and accurately describes the problem and research question. However, the introduction section requires inserting more references.
  • Why did the authors rely on a Web of Science core database only? (as they mentioned in the 2. Materials and Methods Section) although there are many other databases. The authors should state the reason for their choice and add it to the text.
  • Were scientific reports, white papers, case studies, ... etc. also excluded from the search criterion (2. Materials and Methods Section)? Why? The authors should clarify this point.
  • In Table 1, it is recommended that authors add a column for journal fields and a column to start publishing papers.
  • In the Demographic characteristics Section (page4), the authors pointed out that "mobile health applications are more used by the young rather than the senior" but they did not indicate what age classes used these APPs more? (for instance, 20-25, 25-30, … etc.)
  • In 3.3.3. APP design Section (page6), the authors provided a detailed explanation of the factors influencing (such as Functionality, Perceived ease of use and usefulness, ... etc.) the design of mobile health applications. But authors should explain what is the most important and least important influencing factor (or priority of importance such as 1, 2, 3 and 4) in application design in relation to user choice? This point will provide readers with information in determining an area for future investigation.
  • Conclusion and Discussion Section contains valuable information, but this section requires more structuring by identifying suggestions (page9-line10) with explanatory titles. Moreover, what are the directions and future visions of this research?

  • All Figures and Tables are accurately drawn (high resolution) and appropriately invoked in-text.
  • References list: Authors should remove all mistakes in the reference list. For example, references are not arranged correctly in the text, reference [42] appears in-text before references 38-41, reference [1] is not used in-text. References should follow the MDPI-Healthcare style. Some references do not contain enough information such as the reference [14] has not journal name. The reference list needs minor improvement.

Proofreading: This systematic review requires addressing some typographical and grammatical issues to make English writing readable and understandable. For instance, some of the words require "a", "an" and "the" in different places in this review (such as “great” (page1, abstract), “health” (page1-line29, introduction), “stronger” (page4-table2) “positive” (page4-line14), “intention” (page5-line10), … etc.),  remove “on” (page1-line2), use “socio-economic” instead of “socioeconomic” (page1-line15), replace “related with” with “related to” (page4-table2, check whole paper), add “s” to “senior”, “female”, “male”, “student”, … etc. (page4-line8,line10,line11,line16), use “smartphone” instead “smart phone” (check whole paper), replace “generally” with “general” (page5), add “s” to “maintain” (page7), use “significantly” instead of “significant” (page9), … etc. This revision requires careful proofreading to improve the English writing.

Author Response

Point 1: We suggest that the authors should eliminate keywords such as "mobile health applications", "influencing factor" and "systematic review " because these keywords are already found in the article title. It is preferable that they replace them with other words in order to expand the reach of the search.

Response 1:  We replace keywords "mobile health applications", "influencing factor" and "systematic review " with keywords "individual influencing factors", "social influencing factors" and “APP design influencing factors” respectively. (page1-Keywords)

Point 2: The abstract is appropriately written, structured, complete and independent as well as it is correctly describing the title of the paper. However, the authors must define what the acronym "APP" (page1-line7) indicates before using it.

Response 2: It is revised with annotation "application (app or APP) ". (page1-line7)

Point 3: The introduction section provides appropriate background on the topic and accurately describes the problem and research question. However, the introduction section requires inserting more references.

Response 3: The following contents are added to the introduction. (page2)

In view of this problem, this paper systematically reviews the relevant research, and finds that although the research results are abundant, most of them focus on the empirical research on the influencing factors of users’ acceptance and use of mobile health apps, including the research on different groups of people, such as the determinants of adoption of mobile health apps among college students [3], the analysis on the mobile health app possession and related sociodemographic factors among Hong Kong citizens [4], the factors associated with mobile health apps use in cardiovascular diseases or diabetes patients[5]; the research on mobile health apps with different functions, such as the acceptability of mental health apps[6], the opportunities and obstacles for mobile health apps use for cancer care[7], the relationship between the characteristics of mobile health apps related to maternal and infant health and uses’ intent to use[8]; and the research on a certain influencing factor, such as the effect of whether sending a push notification on engagement in mobile health app[2], the relationship between health intentions and mobile health app ownership[9], the obstacles and facilitators of using mobile health apps from the perspective of security[10]. In addition, previous reviews mainly focused on a certain type of mobile health apps (e.g. for patients with rheumatic and musculoskeletal diseases [11], for chronic arthritis patients [12]). To the best of our knowledge, no study has systematically reviewed the influencing factors of users’ acceptance and use behavior of mobile health apps, which is not enough to reflect the general situation of current research. Therefore, this paper makes a comprehensive review to comb and analyse the influencing factors from multiple dimensions, in order to grasp the current situation and trend of relevant research, and provide a reference for the follow-up research.

Point 4: Why did the authors rely on a Web of Science core database only? (as they mentioned in the 2. Materials and Methods Section) although there are many other databases. The authors should state the reason for their choice and add it to the text.

 Response 4: The following content is added in the 2. Materials and Methods Section. (page2)

According to Bradford's Law, most of the key research is published in core international journals. Therefore, Web of Science core database is taken as the data source in this paper.

In addition, we add the “5. limitation” section in the article. (page11)

This review was restricted to the article literature in the Web of Science core database, and does not include other types of literature (such as scientific reports, white papers, etc.) or other literature databases (such as PubMed, Scopus, etc.), thus the research results may not have captured all possible influencing factors of mobile health applications. Non-English articles are also excluded, which limits the generalizability of our findings.

Point 5: Were scientific reports, white papers, case studies, ... etc. also excluded from the search criterion (2. Materials and Methods Section)? Why? The authors should clarify this point.

Response 5: The reason is the same as Response 4.  We initially searched for all types of literature and found that the most valuable literature was journals. Thus, we add the “5. limitation” section in the article. (page11)

This review was restricted to the article literature in the Web of Science core database, and does not include other types of literature (such as scientific reports, white papers, etc.) or other literature databases (such as PubMed, Scopus, etc.), thus the research results may not have captured all possible influencing factors of mobile health applications. Non-English articles are also excluded, which limits the generalizability of our findings.

Point 6: In Table 1, it is recommended that authors add a column for journal fields and a column to start publishing papers.

Response 6: We have revised table 1 as follows. (page4)

Table 1. Journal distribution.

Journals

Number of studies

Journal fields

Start publishing papers

JMIR mHealth and uHealth

7

Health Care Sciences & Services

2014

International Journal of Medical Informatics

6

Health Care Sciences & Services, Medical Informatics, Computer Science

2009

Journal of Medical Internet Research

5

Health Care Sciences & Services, Medical Informatics

2014

BMC Medical Informatics and Decision Making

4

Medical Informatics

2013

PLOS One

4

Comprehensive

2016

Telemedicine and E-Health

3

Health Care Sciences & Services

2015

Health Informatics Journal

3

Health Care Sciences & Services

2016

Digital Health

3

Health Care Sciences & Services

2018

Journal of Cancer Education

3

Medicine Education

2018

Point 7: In the Demographic characteristics Section (page4), the authors pointed out that "mobile health applications are more used by the young rather than the senior" but they did not indicate what age classes used these APPs more? (for instance, 20-25, 25-30, … etc.)

Response 7: The article is amended as follows. (page5-6)

First of all, use behavior partly depends on age, as mobile health applications are more used by the young (although the definitions for the age range of young people in various literatures are different, most of them are under 35 years old[15][16][17][18]) rather than the seniors, especially those over age 70, who are less comfortable using their phone applications[15].

 Point 8: In 3.3.3. APP design Section (page6), the authors provided a detailed explanation of the factors influencing (such as Functionality, Perceived ease of use and usefulness, ... etc.) the design of mobile health applications. But authors should explain what is the most important and least important influencing factor (or priority of importance such as 1, 2, 3 and 4) in application design in relation to user choice? This point will provide readers with information in determining an area for future investigation.

Response 8: The article is amended as follows.

  1. Add the Figure 3, we draw a spider plot according to the frequency of each influencing factor in the relevant literature (as shown in Figure 3). (page 5)
  2. In Table 2, Table 3, Table 4, the “Influencing factor” column is added “Number of related papers”, to show the different importance of the influencing factors. (page 5,7,8)
  3. We added Conclusion content as following. (page 12)

As for APP design, the research findings are relatively rich, including functionality, perceived ease of use and usefulness, security and cost. Firstly, the Technology Acceptance Model is widely used in studying the factors that affect users’ behavior, in which perceived ease of use and usefulness are the two main factors, which appear most frequently in the relevant literature. Secondly, functionality is the focus of researchers, in which the information quality is an important indicator, while core functions (e.g., reminder, notification, incentive, follow-up and goal setting), personalization and gamification also attract the attention of researchers. Thirdly, the importance of security can not be ignored. Finally, research in cost provides a new perspective for application designers.

Point 9: Conclusion and Discussion Section contains valuable information, but this section requires more structuring by identifying suggestions (page9-line10) with explanatory titles. Moreover, what are the directions and future visions of this research?

Response 9: Conclusion and Discussion Section is divided into two sections. (page10-12)

  1. Discussion

4.1 Considering the demographic factors that affect users’ behavior

4.2 Strengthening the publicity and system management of mobile health apps

4.3 Strengthening the security and privacy measures of mobile health apps

  1. Conclusion

The following content is added in the conclusion. (page12)

As the user's intention and behavior play a key role in the innovation and development of mobile health apps, it is bound to be favored by more researchers in the follow-up research. In the future, it may focus on the influencing factors in different application scenarios.

Point 10: References list: Authors should remove all mistakes in the reference list. For example, references are not arranged correctly in the text, reference [42] appears in-text before references 38-41, reference [1] is not used in-text. References should follow the MDPI-Healthcare style. Some references do not contain enough information such as the reference [14] has not journal name. The reference list needs minor improvement.

Response 10: The format of references was revised according to the MDPI-Healthcare style.

Point 11: Proofreading: This systematic review requires addressing some typographical and grammatical issues to make English writing readable and understandable. For instance, some of the words require "a", "an" and "the" in different places in this review (such as “great” (page1, abstract), “health” (page1-line29, introduction), “stronger” (page4-table2) “positive” (page4-line14), “intention” (page5-line10), … etc.),  remove “on” (page1-line2), use “socio-economic” instead of “socioeconomic” (page1-line15), replace “related with” with “related to” (page4-table2, check whole paper), add “s” to “senior”, “female”, “male”, “student”, … etc. (page4-line8,line10,line11,line16), use “smartphone” instead “smart phone” (check whole paper), replace “generally” with “general” (page5), add “s” to “maintain” (page7), use “significantly” instead of “significant” (page9), … etc. This revision requires careful proofreading to improve the English writing.

Response 11: We corrected the grammatical mistakes in the article.

Reviewer 2 Report

This systematic review details factors influencing the acceptance and usage of mobile applications in the healthcare domain. The review is well written, and the suggestions derived from the study can drive guidelines for application-design towards improved user acceptance. 

Some of my comments to improve this review:

  • How did the authors assess the quality of the articles considered for this study? It must be clearly stated in the study. 
  • Addition of visual infographics on factors influencing in the form of spider plot is recommended. 
  • There is a plethora of studies and reviews on "acceptance of mobile health apps." The authors' must discuss what novel conclusions this study brings in irrespective of existing literature. 

Author Response

Point 1: How did the authors assess the quality of the articles considered for this study? It must be clearly stated in the study.

Response 1:  In the Materials and Methods(page2) section, the following contents are added.(page 2)

The selection process was conducted in accordance with PRISMA-ScR (Preferred Reporting Items for Systematic Reviews and Meta Analyses extension for Scoping Reviews)[13].

Point 2: Addition of visual infographics on factors influencing in the form of spider plot is recommended.

Response 2: Add the Figure 3, we draw a spider plot according to the frequency of each influencing factor in the relevant literature (as shown in Figure 3).(page 5)

Point 3: There is a plethora of studies and reviews on "acceptance of mobile health apps." The authors' must discuss what novel conclusions this study brings in irrespective of existing literature.

Response 3: The following contents are added to the introduction to explain this article is different from other studies.(page 2) .

  1. introduction.

In view of this problem, this paper systematically reviews the relevant research, and finds that although the research results are abundant, most of them focus on the empirical research on the influencing factors of users’ acceptance and use of mobile health apps, including the research on different groups of people, such as the determinants of adoption of mobile health apps among college students [3], the analysis on the mobile health app possession and related sociodemographic factors among Hong Kong citizens [4], the factors associated with mobile health apps use in cardiovascular diseases or diabetes patients[5]; the research on mobile health apps with different functions, such as the acceptability of mental health apps[6], the opportunities and obstacles for mobile health apps use for cancer care[7], the relationship between the characteristics of mobile health apps related to maternal and infant health and uses’ intent to use[8]; and the research on a certain influencing factor, such as the effect of whether sending a push notification on engagement in mobile health app[2], the relationship between health intentions and mobile health app ownership[9], the obstacles and facilitators of using mobile health apps from the perspective of security[10]. In addition, previous reviews mainly focused on a certain type of mobile health apps (e.g. for patients with rheumatic and musculoskeletal diseases [11], for chronic arthritis patients [12]). To the best of our knowledge, no study has systematically reviewed the influencing factors of users’ acceptance and use behavior of mobile health apps, which is not enough to reflect the general situation of current research. Therefore, this paper makes a comprehensive review to comb and analyse the influencing factors from multiple dimensions, in order to grasp the current situation and trend of relevant research, and provide a reference for the follow-up research.

In the conclusion section, a novel conclusion is given.(page 11)

  1. Conclusion

Previous studies mainly focused on finding and verifying the factors that affect the users’ acceptance and use behavior of mobile health applications. Different from previous researches, this paper summarizes 10 influencing factors and 24 related variables from three dimensions of individual, society and APP design, analyzes the importance of different factors according to the frequency of influencing factors in the relevant literature, and systematically reviews the influencing factors of users’ acceptance and use behavior of mobile health apps.

Reviewer 3 Report

This is an interesting review of an increasing attention area. There is a lot of information right now in this line, the authors identified the principals research of the area, but probably they condensed too much of the information of the paper. Probably extending this information would be beneficial for the final manuscript, especially in sections APP design, Social dimension, and Individual dimension. A final practical application point would also improve the quality of the paper. 

INTRODUCTION

The introduction must be improved to provide sufficient background information for readers to understand the resear problem.

Include a concise study objective.

METHODS

The experimental approach is appropriate for the aim of the study.

This section is well described and allows to replicate the study.

RESULTS

Results paragraph include more relevant and extended data.

All of the tables include specific, well-developed statistic. Revise format of tables to be in line with journal style.

DISCUSSION

The interpretations of the data considered are consistent.

Explain the implications of results obtained and their practical applications

Provide a concise conclusion that respond the study aims.

LITERATURE CITED

The literature cited is relevant to the study. Revise format to be in line with the text

SIGNIFICANCE AND NOVELTY

As it stands, the results are novel and important enough for this journal.

Author Response

Point 1: INTRODUCTION

The introduction must be improved to provide sufficient background information for readers to understand the resear problem. Include a concise study objective.

Response 1:  The following contents are added to the introduction. (page2)

In view of this problem, this paper systematically reviews the relevant research, and finds that although the research results are abundant, most of them focus on the empirical research on the influencing factors of users’ acceptance and use of mobile health apps, including the research on different groups of people, such as the determinants of adoption of mobile health apps among college students [3], the analysis on the mobile health app possession and related sociodemographic factors among Hong Kong citizens [4], the factors associated with mobile health apps use in cardiovascular diseases or diabetes patients[5]; the research on mobile health apps with different functions, such as the acceptability of mental health apps[6], the opportunities and obstacles for mobile health apps use for cancer care[7], the relationship between the characteristics of mobile health apps related to maternal and infant health and uses’ intent to use[8]; and the research on a certain influencing factor, such as the effect of whether sending a push notification on engagement in mobile health app[2], the relationship between health intentions and mobile health app ownership[9], the obstacles and facilitators of using mobile health apps from the perspective of security[10]. In addition, previous reviews mainly focused on a certain type of mobile health apps (e.g. for patients with rheumatic and musculoskeletal diseases [11], for chronic arthritis patients [12]). To the best of our knowledge, no study has systematically reviewed the influencing factors of users’ acceptance and use behavior of mobile health apps, which is not enough to reflect the general situation of current research. Therefore, this paper makes a comprehensive review to comb and analyse the influencing factors from multiple dimensions, in order to grasp the current situation and trend of relevant research, and provide a reference for the follow-up research.

Point 2: All of the tables include specific, well-developed statistic. Revise format of tables to be in line with journal style.

Response 2: The format of article was revised according to  the MDPI-Healthcare style.

Point 3: LITERATURE CITED

The literature cited is relevant to the study. Revise format to be in line with the text

Response 3: The format of references was revised according to  the MDPI-Healthcare style.
